# Neuropsychological and Neurophysiological Mechanisms behind Flickering Light Stimulus Processing

**DOI:** 10.3390/biology11121720

**Published:** 2022-11-28

**Authors:** Natalia D. Mankowska, Malgorzata Grzywinska, Pawel J. Winklewski, Anna B. Marcinkowska

**Affiliations:** 1Applied Cognitive Neuroscience Lab, Department of Human Physiology, Medical University of Gdansk, 80-210 Gdansk, Poland; 2Neuroinformatics and Artificial Intelligence Lab, Department of Human Physiology, Medical University of Gdansk, 80-210 Gdansk, Poland; 3Department of Human Physiology, Medical University of Gdansk, 80-210 Gdansk, Poland; 42nd Department of Radiology, Medical University of Gdansk, 80-210 Gdansk, Poland; 5Institute of Health Sciences, Pomeranian University in Slupsk, 76-200 Slupsk, Poland

**Keywords:** flickering light, critical flicker fusion frequency, visual evoked potentials, electroencephalography, magnetic resonance imaging, visual system

## Abstract

**Simple Summary:**

Flickering light is used in research in many different fields. Despite growing interest in the subject, there is still little known about its effects on the brain. The researchers used flickering light in different variations, so it is important to analyse how these modifications may affect the obtained results. This review relates to both neurophysiological and anatomical aspects of this topic, including processing visual stimuli in the brain, especially colour and motion. Since the results of flickering light-based tests (e.g., flicker test) have been linked to arousal levels in the literature, this review also describes this topic, along with attentional processes and detection of signals in the visual field.

**Abstract:**

The aim of this review is to summarise current knowledge about flickering light and the underlying processes that occur during its processing in the brain. Despite the growing interest in the topic of flickering light, its clinical applications are still not well understood. Studies using EEG indicate an appearing synchronisation of brain wave frequencies with the frequency of flickering light, and hopefully, it could be used in memory therapy, among other applications. Some researchers have focused on using the flicker test as an indicator of arousal, which may be useful in clinical studies if the background for such a relationship is described. Since flicker testing has a risk of inducing epileptic seizures, however, every effort must be made to avoid high-risk combinations, which include, for example, red-blue light flashing at 15 Hz. Future research should focus on the usage of neuroimaging methods to describe the specific neuropsychological and neurophysiological processes occurring in the brain during the processing of flickering light so that its clinical utility can be preliminarily determined and randomised clinical trials can be initiated to test existing reports.

## 1. Introduction

In the last hundred years, flickering light has attracted increasingly more attention from researchers around the world. From 1933 to 2021, more than 6000 related articles appeared in the PubMed database alone. Critical flicker fusion frequency (CFF or CFFF), as a frequency at which flickering light stops being visible and starts being perceived as a steady light, has been used in various fields of research and various groups or species. Research has focused on dementia [1,2,3], visual impairment [4], cognitive functioning [5,6,7,8] and divers [9,10,11,12,13,14], among other topics. 

Despite growing interest in the topic, the neurophysiological and neuropsychological mechanisms behind the processing of flickering light stimuli are still not fully understood. This review aims to integrate existing knowledge and determine a valid basis for the mentioned processes that occur during observing light and detecting its blinking or stability. This knowledge could contribute to the development of new clinical and research interventions and diagnostic procedures.

## 2. Flickering Light

The CFF threshold in humans is estimated to be 50–90 Hz [15,16], although some researchers have reported that distinguishing between flickering and steady light can also occur at higher frequencies, i.e., 500 Hz [17], which they explain by the emergence of the unconscious, rapid, saccadic eye movements that are made along the edges of the projected image. The results of previous studies are not consistent with each other, as some researchers have reported that eye movements can positively increase the frequency needed to perceive a stable image [18,19], while others reported no such regularity [15,20].

Distinguishing between flickering and steady light is dependent on the factors mentioned in the review article by Mankowska et al. [21]: (1) the frequency of modulation, (2) the amplitude of modulation, (3) the average illumination intensity, (4) the position on the retina at which the stimulus occurs, (5) the wavelength or colour of the light-emitting diode, (6) the intensity of ambient light, or (7) the viewing distance and (8) size of the stimulus. According to Bullough and colleagues [22], the combination of light flickering at 100 Hz or more with a stable, stationary background is associated with: (1) the perception of that light as a more constant and (2) less stroboscopic effect. However, there have been reports that a stroboscopic effect can occur even with the light flickering at more than 1000 Hz if objects in the rest of the visual field are moving [23,24].

Rider et al. [25] summarised results from research conducted over the last twenty years and concluded that CFF has limited value as a clinical measurement because while it contributes to a better understanding of the visual system’s overall gain, it does not provide much information on processing speed. The CFF is dependent on both gain and processing speed, so measurements need to take several intensity levels into account to acquire information about this dependence. It would be beneficial to follow damage in different parts of the visual pathway since it causes diverse reactions.

## 3. Visual System and Visual Information Processing

Flickering light is a visual stimulus of relatively high complexity, as part of its processing by the visual system, not only the size is analysed, but also the motion (flickering), colour, the size of the stimulus and spatial relations (the arrangement of the stimulus and whether a single element or a more complex arrangement, such as a checkerboard, is presented). Thus, it is necessary to understand how the visual system processes information of different types.

Light comes into the eye through the pupil, before being focused by the lens and cornea, through which it is projected onto the retina (Figure 1). The retina contains photoreceptors that convert photon energy into neuronal activity [26]. The left half of the retina processes information from the right visual field and the right from the left visual field. The image on the retina is upside down, just like in a camera. The retinas of vertebrates have two types of receptors, with about 5 million cones and 120 million rods [27]. Rods are responsible for scotopic (night) vision, while cones relate to photopic vision (when there is plenty of light) and play a significant role in colour perception. Both have visual pigments that emit energy when exposed to light. Rhodopsin is a photosensitive pigment of rods and absorbs wavelengths of about 400–600 nm to create action potential and transmit information from the retina to the brain. Cones have four types of pigments, absorbing short, medium and long wavelengths [27,28]. 

Information from the retina is transmitted to the visual cortex through the lateral geniculate nucleus, after receiving it from ganglion cells, which corresponds to a receptive field where the transmitted light causes a change during the electrical activity. There are three types of responses from ganglion cells: on, off and on-off. They show, respectively: (1) increased activity occurring only during light stimulation, (2) inhibited activity when the light occurs and acceleration after turning the light off and (3) increased activity immediately after turning the light on and off [30,31].

The most common types of ganglion cells are magno and parvo. Magno cells process information about luminance, motion, stereopsis and depth of objects since they receive information from all cones and can quickly conduct this information (about 40 m/s). Parvo cells are slower (20 m/s), more numerous than magno (80% of total compared to 10% of magno cells) and more sensitive to colours and stationary patterns such as the material and form of objects. This differentiation is reflected in the connection with visual information processing pathways: while information from magno cells is processed by the “where” pathway (dorsal stream), the “what” pathway (ventral stream) processes information from parvo cells [32]. Figure 2 shows a course of both pathways [33,34]. The V1–V6 indicates brain regions, as described in Table 1 and shown in Figure 3.

Some research showed that reduced magnocellular sensitivity is associated with problems in rapid visual processing and thus with poor reading and dyslexia, while there is no such a correlation in parvocellular ganglion cells, in neither better nor poor readers [39,40]. This mechanism is probably associated with visual attention and eye fixation on words. As reported by Stein [40], reading in these individuals may be improved by boosting M-performance by using yellow filters or training eye fixation. Graves et al. [41] indicated that adults with reading disabilities had problems with localising small, briefly presented visual stimuli. Moreover, Peters and colleagues [42] found that dyslexic individuals compared to neurotypicals have a lower ability to detect flickering at high temporal frequencies, but they also emphasise that this difference is not sufficient to discriminate between groups. 

Foxe et al. [43], in their experiment with the use of electroencephalography (EEG) and visual evoked potentials (VEPs), showed that the C1 component occurred in the central parieto-occipital area, which is only seen when parvo cells are excited (with chromatic isoluminant stimuli). It has been possible to extract other meaningful information from event-related potentials as well. Some researchers showed that the timing of display of a target stimulus among some non-target stimuli could be identified by a single trial [43,44]. MRI studies would be useful to explore this topic further.

Experiments with positron emission tomography (PET) scans revealed that the lateral part of Brodmann’s area (BA) 19 is responsible for movement perception in the contralateral visual hemifield. Area 7, more widely known as the posterior parietal cortex, receives main projections from area 19 and is involved in stereopsis (three-dimensional vision). This area is connected to the pulvinar and projects to the ipsilateral frontal eye field and premotor cortex (PMC) via the superior longitudinal fasciculus [36]. The electrical stimulation of BA 17, 18 and 19 may lead to the production of visual sensations (usually in the contralateral field), often described as flickering lights, stars, lines and spots, among others [45].

Berryhill and Olson [46] performed case-study experiments with two patients with bilateral parietal lobe damage (revealed in MRI) and showed that this part of the brain may be relevant to performing tasks using visual working memory, especially in the retrieval of this kind of material. These findings were replicated in a different group of seven patients with right parietal lobe damage, also confirmed by MRI [47]. They conducted a few experiments based on remembering items to compare performance in tasks with cued recall and old/new recognition. This choice was made after consideration of the research and review of Marshuetz and colleagues [48,49]; they found order working memory-related activity in functional MRI in bilateral parietal regions (BA 7 and 40) which was deemed important for mediation storage operations, e.g., tracking the temporal spacing between items. 

Working memory, like some aspects of attention, is considered to belong to executive functions, which may be relevant in analysing the mechanisms of flickering light. The parietal cortex is thought to be activated in tasks requiring sequencing, especially motor sequencing. Gonzalez and Burke [50] examined 12 young participants with functional MRI and an eye tracker to observe brain activity during the visual learning of simple and more complex trajectories. They found the following: (1) the activation of memory and attention-related areas (BA10 and dorsolateral prefrontal cortex) during the shorter sequences and (2) activation within areas that corresponded to the pre-motor (BA6) and motor (BA4) cortex during longer sequences as an effect of movement planning and the preparation of movement. In the comparison of both short and long tasks, there were no differences within the parietal cortex; however, the higher activation was observed in BA40 for predictive versus random conditions. 

## 4. Colour Vision

In the literature, flickering stimuli are presented in different colours. As is known, particular rods and cones process slightly different wavelengths of visible light. Interactions between different colours are important, among other things, by their ability to induce epileptic seizures, as described in Section 10.

Humans can perceive light with a wavelength of approximately 350–700 nm [51]. The shortest wavelengths are responsible for the impression of the violet colour and the longest of red (see Figure 4). Perception relies on the activity of multiple neurons because any one of them can simultaneously transmit information about brightness and colour [34]. 

## 5. Motion Processing

As mentioned above, the brain structures responsible for analysing motion are also involved in processing flickering stimuli. The most important area associated with the perception of motion stimuli, as described by researchers so far, is area V5 (medial temporal, MT) [52]. Researchers used various neuroimaging and neurostimulation techniques such as PET [53], MRI [54], functional MRI [55,56], transcranial magnetic stimulation (TMS) [57] and magnetoencephalography (MEG). The latter two were used to describe its localisation more specifically: “human V5 is located near the occipito-temporal border in a minor sulcus immediately below the superior temporal sulcus” [54]. Damage to this area results in akinetopsia, firstly described by Zihl et al. [58] and confirmed by Shipp et al. [59] using MRI. 

Area V5 is divided into two subdivisions which allow primates to (1) perceive the motion of objects and (2) analyse object images after making a head movement [60]. Ashida and Osaka [61] described motion aftereffect (MAE) which occurs after gazing at a stimulus moving in some direction for a long period of time; a later viewed stationary scene seems to move then in the opposite direction. There are two types of MAE: (1) static, where the maximum aftereffect is reached when the spatial frequency of both adapting and test stimuli are equals because of reflection of Fourier power and (2) flicker, which follows the pattern direction. Researchers concluded that flicker MAE is probably more dependent on velocity than on temporal frequency, unlike static MAE, which shows spatial frequency selectivity. It is highly possible that these types of MAE are processed in different stages of the visual pathway; while static is analysed in V1, flicker MAE needs the integration of more information entering V5.

## 6. Flickering Light in Neuroimaging Studies

EEG is a non-invasive method of studying the bioelectrical activity of the brain that has been used successfully in clinical studies of patients with epilepsy, sleep disorders and other neurological problems, diseases and psychiatric disorders, among others [62]. It is also useful in studies of healthy subjects, for example, to observe brain activity at rest or when focusing attention on cognitively demanding tasks. Each state of brain activity is reflected in different frequencies of brain waves. The electrodes used in research vary in number, depending on how accurate the information needed is. 

Human visual perception tends to change in synchronisation with neural oscillations, especially in the delta, theta and alpha frequencies in the EEG. Some researchers focus on alpha oscillations, as variations in individual alpha frequency are the most prominent and correlate with discrimination between stimuli presented in the same location [63]. For example, Spaak et al. [64] showed that 10 Hz flicker caused changes in entraining that persisted for approximately 300 ms after ceasing stimulus observation. Gray and Emmanouil [63] tested whether rhythmic visual stimulation at lower and higher alpha frequencies (8.3 and 12.5 Hz) may have an impact on differentiating one from two flashes. They found that individual alpha frequency only reflects neural processes which are independent of environmental factors and regulate perception internally, but they emphasised that further research is needed to unambiguously explain this phenomenon.

According to Herrmann [65], external stimulation can drive the visual cortex up to 50 Hz. Moreover, in the steady state VEP (SSVEP), the resonance phenomena are visible, which indicates a stronger response on some frequencies (10, 20, 40 and 80 Hz), probably because of the preferred oscillation frequency, resulting from axonal connections between neurons in a specific system. In research with patients with multiple sclerosis, Salmi [66] found that the flicker test may have additional diagnostic value in optic neuritis compared to VEP from O1 and O2 electrodes. Whether there were differences in the CFF threshold between both patients’ eyes was examined. Interocular discrepancies were taken as an indicator of abnormal functioning of the visual pathway in some optic neuritis cases. The abnormal CFF threshold was found in 46% of the group of sixty-three patients and it was significantly lower than in the controls, which is consistent with previous research in larger groups of patients (48–78% of abnormal results, depending on the method used) [67]. For comparison, VEP values were pathological in 69% of cases, compared to previous results—in 81–95% of definite and 26–50% of possible cases of multiple sclerosis [68,69,70].

Another helpful method for studying brain activity is functional magnetic resonance imaging (fMRI), which uses analysis of the level of oxygen consumption by brain tissue. Since its inception in the 1990s, several hundred thousand scientific papers have been written, primarily in neuroscience and psychiatry. Its applications are wide-ranging and include, among others, presurgical planning, monitoring the effectiveness of therapy, or comparing patterns of brain activity in different clinical groups [71]. In addition, it allows the mapping of brain activity and interactions between activated areas, which in research on flickering light makes it possible to unveil structures involved in its processing.

In 2006, Carmel and colleagues analysed brain activity when observing a flickering point of light, as examined by fMRI [72]. They noted increased activity in the cortex of both frontal lobes and the left parietal cortex. These areas have previously been reported to be associated with awareness during non-temporal tasks (with the perception of intervals), but the results of the mentioned study suggest that they may nevertheless play a role in these types of tasks as well. These results are consistent with theories of visual awareness that arise from the interplay of multiple networks within the brain. However, whether frontoparietal areas play a role in the detection of flickering or whether they are activated by the performance of the flicker detection task itself cannot be determined from fMRI studies.

In a functional MRI study, Zafiris and colleagues [73] analysed neuronal mechanisms of processing flickering light in patients with liver cirrhosis. They used part of the computerised neuropsychological test battery (Vienna Test System) and the range of flicker frequency used in their paradigm was 25–50 Hz, slowly decreasing from higher values by 0.5 Hz. They tested 9 patients and 10 control subjects and found activation differences in the right inferior parietal cortex, the parietooccipital cortex (cuneus), the anterior cingulate cortex, the intraparietal sulcus, the medial temporal lobe, the thalamus, prefrontal polar cortex and striate and extrastriate visual cortices. Moreover, the functional MRI signal in patients was reduced in the inferior parietal cortex compared to healthy controls, which may indicate the importance of this area for visual attention. Interestingly, this finding is correlated with enhanced signals in the temporal pole (see Table 1 for its functions in visual processing), probably as the compensatory mechanism. As researchers emphasised, according to Butterworth [74], chronic liver failure may result in the production of brain cells of Alzheimer’s type II; thus, a better understanding of these matters would benefit researchers, clinicians and, most importantly, patients.

In their research with a comparison of EEG and functional MRI responses, Mullinger et al. [75] recorded data from 17 subjects (mean age: 26 years) and used a black-white checkerboard in two conditions: static and flickering at 3 Hz (in functional MRI). The aim of their study was to differentiate whether weakening of the functional MRI signal (in response to 8 Hz flicker) after the cessation of stimulus observation is due to neuronal or vascular changes. Their results provide evidence that the origin of this phase response is neuronal as the amplitudes of functional MRI signal and cerebral blood flow signals were dependent on the post-stimulus power of the occipital alpha EEG neuronal activity. When the EEG powers were the highest, the lowest activity in functional MRI was observed in the contralateral visual cortex. Mullinger and colleagues pointed out that a more accurate analysis could contribute to a better understanding of cognitive dysfunctions and neurological diseases. There is limited evidence that the visual cortex responds to a flickering stimulus, not only when the flickering is perceivable, but also when the stimulus seem to be steady. Few healthy subjects participated in the study by Jiang et al. [76] and observed a 5 and 30 Hz full contrast chromatic flicker. When they perceived light flickering at lower frequencies, they showed a large functional MRI signal in the visual cortex which also persisted at higher stimulus frequencies.

To the best of our knowledge, there is a lack of studies comprehensively describing the neuropsychological and neurophysiological mechanisms of flickering light processing in the brain. Mentioned studies showed that processing this type of stimuli is complex and involves many brain regions—not only the visual cortex is involved in processing flickering light, but also frontal areas associated with attention and decision-making processes. Examples of conducted studies show that flickering light can have diagnostic and therapeutic potential, but at this point, there is no robust explanation of the principles of these interactions.

## 7. Visual Attention

When two stimuli are presented concurrently, competition between them appears. It is a mechanism of selective attention, described by Desimone and Duncan [77] as a “biased competition model” which assumes that these stimuli would suppress the response from neurons; this was confirmed by Fuchs et al. [78] when they showed that SSVEP amplitude decreased in these conditions. The amplitude of SSVEPs may be increased by paying attention to flickering stimuli [79,80,81]. Any flicker frequency is responsible for activating different cortical networks that are responding to specific frequencies, for example, flickering at lower frequencies (to 10 Hz) activates a global cortical network that is distinguishable from other regions [80,81].

To investigate these competitive interactions and attention, the frequency-tagging method might be used. Its main concept is that different frequencies are assigned to each stimulus to elicit SSVEP in EEG. As the SSVEP amplitude changes in response to the perceived frequency, it is possible to separate them in analysis [82]. 

De Lissa et al. [83] used the frequency tagging method to assess visual attention in the smooth-pursuit paradigm, where 17 participants had to follow a moving target and react when it reached the goal area (1st paradigm) or follow this target and switch their attention when the second one, flickering at 30 Hz, occurs and follow it until it reaches the goal area (2nd paradigm). Researchers found that SSVEP power decreased early and rapidly when the individuals had to divide their attention. These findings are consistent with Kahnemann’s and Lavie’s theories about shared attentional resources for covert and overt attention [84,85], which shows that not only the current task is relevant but also surroundings to which, as needed, attention resources are shifting. In a laboratory environment, where task performance is strictly controlled by researchers, this naturally has implications for the ability to compare patterns of neuronal activity. In everyday life, when a variety of stimuli arrive, affecting different modalities, these relationships become less obvious. It is therefore important to consider how performance is affected without this control.

## 8. Psychophysics of Stimuli Detection 

This section attempts to describe how various stimuli (especially flickering light), more or less relevant to the individual’s current expectations, are processed by this individual. Basically, the perception of flickering light can be considered in the context of bottom-up and top-down processing theories. In bottom-up processing, incoming stimulus information comes from sensory data and is transferred to the higher levels of the brain for further analysis. Top-down processing is reliant on an individual’s knowledge and prior experiences to analyse current stimuli. Currently, research shows that both processes support orienting to stimuli and the human dorsal frontoparietal network is activated in its searching and detecting. It interacts with the extrastriate cortex and the brain activity occurs as soon as the search field is seen and persists until the target is detected, which was confirmed with neuroimaging methods. The ventral parietal network might be activated in tasks with the detection of low-frequency stimuli at an expected location or when the stimuli were fixed at the centre of gaze (see review by Corbetta & Shulman [86]). These authors summarised the areas of the brain that are involved in performing tasks of the first kind, which are the right temporoparietal junction and ventral frontal cortex (including the inferior frontal gyrus, middle frontal gyrus and frontal operculum). It is worth noting that lateralisation to the right hemisphere was observed. Referring to the mentioned review, it could therefore be hypothesized that similar brain regions will show their activity when observing a flickering light and when making decisions about its properties.

According to signal detection theory (or sensory decision theory), making statements about perception is based on two parameters: the level of observer’s sensitivity and the process of decision making [87]. Any sensory message can be described in two parts: signal and noise. The signal is a stimulus that one attempts to detect and the noise is also known as a distractor, which may be a physiological activity, thought, requirement of the experimenter, noise, etc. When an observer must decide whether the stimulus is perceived or not, any response can be assigned to one of four categories: hit, miss, false alarm or correct rejection (Table 2). The number of responses depends on the observer’s decision-making style—conservative ones would commit fewer false alarms and hits compared to lax observers [27].

When observers must decide whether stimuli are different from each other, another sensory ability may be seen. The minimum intensity difference required for differentiating two stimuli is called the difference threshold [27]. The value of the threshold can be estimated by many methods, but the most reasonable and common [66,88,89,90] in the flicker test seems to be the method of limits. This method is based on the presentation of two series of stimuli: ascending and descending (in flicker test it would be manipulating of frequency value). This strategy allows problems associated with anticipation and habituation errors to be excluded. As Grondin [27] indicates, responses in each sample may vary considerably, so it is necessary to calculate average response values over all samples to estimate the absolute threshold. Thus, based on theoretical models, it seems that the usage of the limit method is reasonable and could allow for a smaller margin of error in flickering light judgement.

Interpretation of tasks using flickering light depends on the protocol used. Flickering stimuli appear in scientific research in different variants—from simpler ones (e.g., a flickering dot on a uniform background or accompanied by a stable reference dot) to more complex ones (e.g., flickering checkerboards). In signal detection theory terms, this complexity (i.e., usually more noise) can translate into difficulties in the decision-making process about signal properties. Therefore, it is essential to remember that comparing brain activity in response to the processing of these different stimuli should be done with extreme caution in concluding.

## 9. Does the Flicker Test Examine Arousal?

There are a number of studies using the flicker test and neuropsychological tests that seem to indicate a connection between the flicker test and arousal [21]. However, comparing CFF and psychological test results does not provide insight into the underlying processes behind these associations.

In previous studies, the flicker test has been used to assess arousal status and visual processing. These aspects have been studied most often using EEG [63,91,92,93,94,95] and less frequently, functional fMRI [73,76,96] or no method other than the flicker test [11,97].

The validation for using the flicker test to assess alertness and arousal is provided, among others, in a study of 10 patients with narcolepsy and 10 age- and sex-matched volunteers conducted by German researchers [97]. The mean age of subjects was 42 years and they performed a test every 15 min for 10 h; narcoleptic patients had higher variability in the level of task performance throughout the day compared to the control group, suggesting that their alertness fluctuated over time, although the CFF threshold differences were not significant between the two groups. 

A study by Ronzhina and colleagues [98] assessed whether simply performing a flicker test affects an individual’s arousal state. The EEG electrodes were arranged according to the international 10–20 system, but the analysis used recordings from three electrodes (Pz, O1, O2) because they did not show blinking artifacts and because they clearly depict alpha wave activity (8–12 Hz), which is characteristic of a relaxed state. The authors reported that performing the flicker test did not affect participant arousal or alertness levels, but it is important to note that the study included only seven participants after sleep deprivation (they slept for 4 h).

Lecca et al. [99] conducted a study in which they assessed the mental fatigue of 30 professional drivers caused by driving a bus. The study collected information from the flicker test and heart rate variability (HRV), among other things. Their results showed a slight decrease in CFF at the end of a 6 h drive only in the condition of assessing the increasing frequency of flickering light (*p* = 0.041). They also found that heart rate decreases over a few hours of driving (in comparison of initial, central and final values).

Thus, the results of the presented research so far are inconclusive and it is hard to answer the question presented in the title of subsection. Although the flicker test is used in studies as a method of assessing arousal, for now it does not appear to be a sensitive and robust method. It would be beneficial to conduct further research to evaluate the use of this test in research on this topic. It is necessary to find out what relationships connect the CFF with other psychometric tests and physiological factors so it can be considered a reliable tool. The purpose of this work was to summarize the current knowledge of flickering light and its effects on the brain, so given the question posed in this section, it is necessary to consider what brain and cognitive processes could reflect the possibilities in the perception of the changing properties of flickering light.

## 10. Flickering Light, Headaches and Epilepsy

Occasionally, light from lamps flickering at a high frequency, i.e., 100–120 Hz, can induce headaches or migraines [100]. Discomfort from visual strain can also result from observing specific stimuli, such as stripes [101,102] or blurred images [103]. It is worth noting that these stimuli elicit a discomfort response often related to the subject’s state (stress level, cognitive functioning, fatigue, light sensitivity, etc.). According to a study by Yoshimoto et al. [104], strong subjective discomfort is caused by light that has excessively strong contrast and flashes at a medium frequency (around 15 Hz), which confirms a previous study published by Lin et al. [105].

Watching flickering lights can induce epileptic seizures, particularly in individuals who have photogenic epilepsy. However, photoparoxysmal response (PPR) can also be induced in people who have never experienced them before. In the literature, cases of epileptic seizures have been reported in children while watching an episode of the television show Pokémon in which an intense blue-red light flashed at a frequency of 15 Hz [106,107]. Later studies confirmed this was due to stimulation of the most numerous, red cones in the retina [108], so the red light, with a wavelength of about 700 nm, might be the most harmful in these conditions [109]. This relationship is not clear, because Binnie and collaborators [110] showed that some individuals might respond similarly to green light if they had greater sensitivity to green cones than to red cones. Furthermore, patients who had seizures during the colour stimulation also had them during stimulation with a black and white pattern and the number of PPRs was greater than in patients who did not respond to any of these, which was probably a sign of increased photosensitivity [111].

The onset of PPRs is derived from the orientation of stimulus patterns, which is due to the activity of neurons of the visual cortex [102]. Therefore, some people might be more or less sensitive to particular directions [111] although this sensitivity may change over time [112]. Based on their research, Parra and colleagues [111] concluded that the photosensitivity is larger during one-colour stimulation compared to alternating colours stimulation; this is because of the different response mechanisms in visual system pathways: (1) magnocellular responses rely on luminance and high frequencies, (2) parvocellular are more sensitive to longer wavelengths (red-green) and (3) koniocellular to blue-yellow as they have shorter wavelengths. Consequently, the Pokémon incident and its confirmation in research suggest that a red-blue alternating pattern at 15 Hz involves more neurons and cortical space compared to a white/single-color stimulus [111].

To minimise the risk of inducing seizures in people who have never had them before, experiments using potential risk factors should consider the findings of Parra and colleagues [111]. They noted that white lights flashing at higher frequencies (above 20 Hz) can induce epileptic seizures, but that for coloured stimuli (e.g., blue and red) the risk is greatest at frequencies around 15 Hz (see Table 3 for summary). Among the coloured stimuli, it appears that the combination of blue and green is the safest (they elicited the fewest epileptiform responses). Moving forward, Takano and his team [113,114] presented results indicating that participants observing blue-green light experienced less discomfort and performed better than those observing traditional white-grey matrices.

The phenomena and relationships described above should guide future research. First, it is worth monitoring the level of the mentioned discomfort and, on this basis, consider planning possible breaks—so its high level does not significantly affect the obtained results. Second, for ethical reasons, one should design experiments to minimize the risk of inducing epileptic seizures. Third, the mentioned studies indicate that the colours of the stimuli used in flickering light studies affect elements of the visual system differently. Therefore, comparing different experimental designs may have limited relevance.

## 11. Future Directions

To the best of our knowledge, no review has yet been written that summarises the neurophysiological and neuropsychological mechanisms underlying the visual processing of flickering light. Therefore, we have brought together the available information and research findings in this review, which may contribute to the implementation of new interventions among patients from different clinical groups.

Current, but rather limited, knowledge suggests that the processing of flickering light is complex and involves numerous brain regions. However, none of the studies we found explicitly describes the mentioned mechanisms with the step-by-step tracing of the activity of different brain regions. There are not many studies that have combined the analysis of flickering light in EEG and functional MRI and such analysis from different neuroimaging methods could provide more reliable, accurate data.

Future research could also explore the use of flickering light in new technologies. One of them is the brain–computer interface (BCI), which allows neurophysiological signals to be used to control external devices or computers. In past studies, BCI has been tested among patients with amyotrophic lateral sclerosis and cervical spinal cord injury [115,116,117,118,119]. Developing this knowledge could allow the development of forms of examination and support for people with disabilities, particularly those with mobility and communication problems for whom limited options are using standard methods.

In 2021, Norton et al. [120] described BCI-based method for assessing colour vision which is independent from subject’s active participation so it could be used among people who may have difficulty responding, e.g., children or people with cognitive and motor impairments. This method uses SSVEP to the identification of metamers by two flickering light sources. Moreover, they suggest the application potential of this method in the industry. Another promising result of the experiments carried out is that the mentioned method may allow differentiating people with and without colour vision deficits, which, however, needs to be confirmed by further studies using other methods, i.e., anomaloscope. The usage of this method can also contribute to understanding the neural basis of colour perception and to designing therapeutic interventions for people with colour perception disorders.

Combining flickering light with virtual reality would also be an interesting research direction. Recently, a study by Moncada et al. [121] was published in which VR with flickering light (1–50 Hz) was used to identify PPR and photic-driving responses in healthy subjects and light-sensitive subjects. Extending this research into clinical trials among a larger group of patients would be beneficial from the perspective of detecting and treating neurological diseases.

## 12. Conclusions

This review brings together information on flickering light, its features processed by the visual system, and its possible applications in neuroimaging and clinical studies. Information on visual stimulus processing in general, as well as its colour and motion aspects are also included. All information has been analysed from neurophysiological and neuropsychological perspectives. 

To summarize the existing knowledge on the described topic: (1) it is known that flickering light is a complex stimulus, processed by many brain structures that work closely together, (2) there are attempts to use flickering light in the assessment of arousal, (3) flickering stimuli can induce epileptic seizures, depending on their colour and flickering frequency, (4) flickering light can cause discomfort and headaches, (5) processing of information about flickering light is dependent on attentional resources and (6) comparative studies using neuroimaging methods are needed to further verify the described information.

## Figures and Tables

**Figure 1 biology-11-01720-f001:**
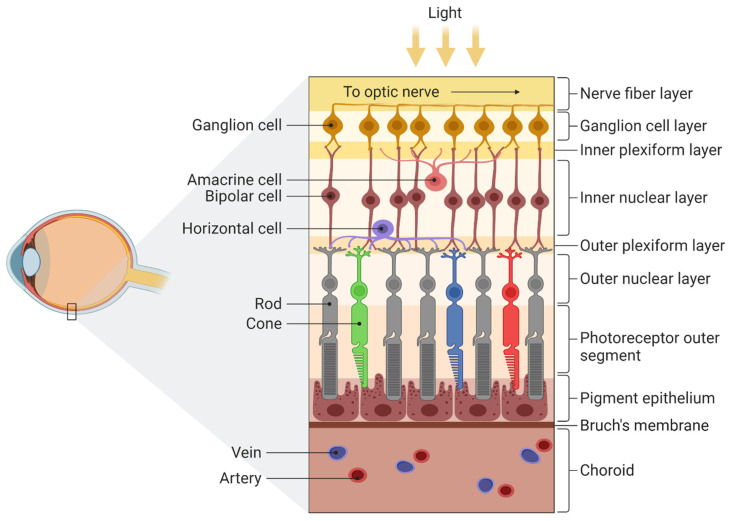
Structure of the retina. There are four main layers: (1) pigment epithelium, (2) the light-sensitive cells such as rods and cones that transmit information to a layer of neurons called (3) bipolar cells which are connected to (4) ganglion cells, located directly under the optic nerve. For more detailed information see Mahabadi & Khalili [29]. Adapted from “Structure of the Retina”, by BioRender.com (2022). Retrieved from https://app.biorender.com/biorender-templates, accessed on 9 September 2022.

**Figure 2 biology-11-01720-f002:**
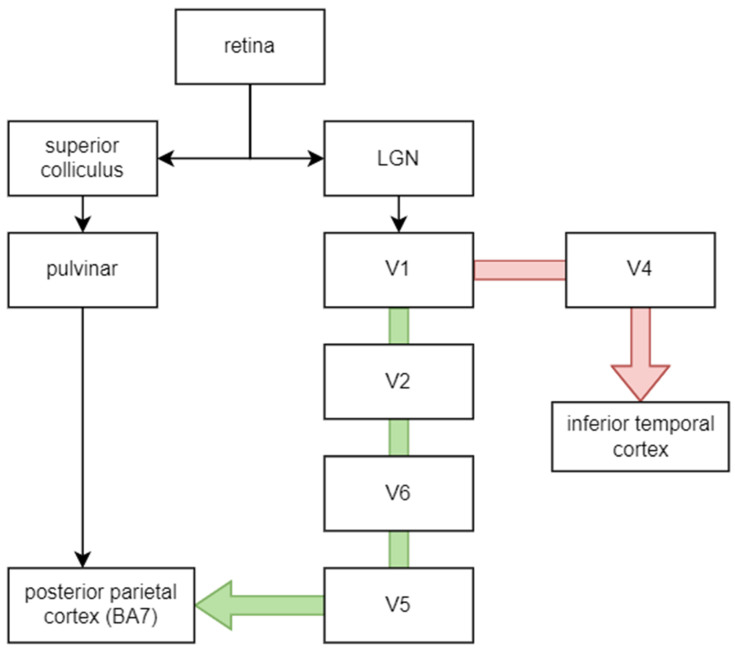
The visual information pathway. The green arrow indicates the dorsal stream, while the red one indicates the ventral stream. LGN—lateral geniculate nucleus.

**Figure 3 biology-11-01720-f003:**
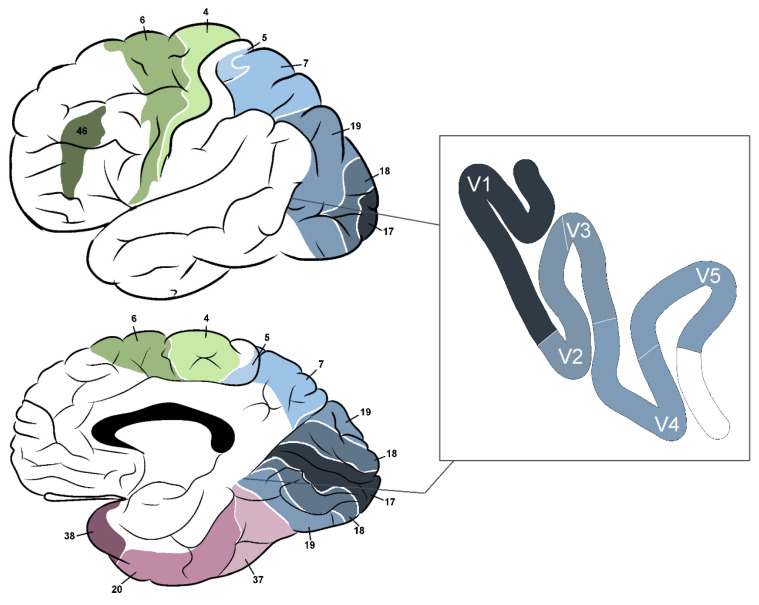
Brodmann’s areas relevant to the visual pathway; 4—primary motor cortex; 5—superior parietal lobule; 6—premotor cortex and supplementary motor area; 7—medial part: precuneus, lateral part: superior parietal lobule; 17—primary visual cortex; 18, 19—secondary visual cortex; 20—inferior temporal gyrus; 37—fusiform gyrus; 38—temporopolar area; 46—dorsolateral prefrontal cortex.

**Figure 4 biology-11-01720-f004:**
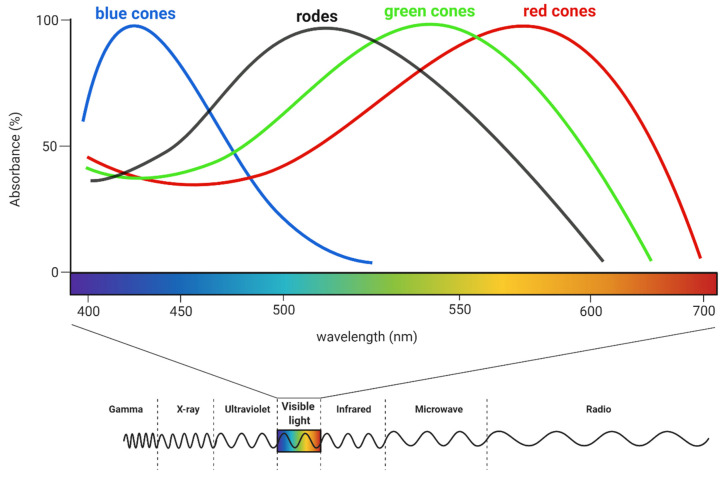
The sensitivity of rods and cones on different light wavelengths. Humans can perceive electromagnetic waves with a length of about 350–700 nm. For comparison, gamma-ray waves have a length less than 10^−12^ m, while radio waves are longer than 1 mm. Created with BioRender.com.

**Table 1 biology-11-01720-t001:** Brain regions associated with visual pathways, their location, function and the most common effects of damage [35,36,37,38].

Brodmann Area	Brain Area		Main Function		Typical Results of Damage
	Functional	Neuroanatomical			
4	Primary motor cortex	Precentral gyrus	Contralateral finger, hand and wrist movement (dorsal) Contralateral lip, tongue, face and mouth movement (lateral) Learning motor sequences Voluntary blinking and inhibition of blinking Horizontal saccadic eye movement Response to touch/observed touch (left) Attention to action (posterior)		Contralateral spastic palsy Contralateral hemiparesis Babinski sign Clonus
5, 7	Secondary sensorimotor cortex	Posterior parietal cortex Medial part: precuneus; lateral part: superior parietal lobule	Visuo-motor coordination Stereopsis Movement perception Saccadic eye movement Working memory Visuospatial memory (right) Tactile localisation (dorsal stream) Visuomotor attention Temporal context recognition (left 7)		Astereognosis Neglect Optic ataxia Apraxia
6	Premotor cortex and SMA	Medial frontal gyrus	Motor sequencing/planning Motor learning (SMA) Movement preparation/imagined movement (rostral SMA) Movement initiation (caudal SMA) Motor imagery (SMA) Horizontal saccadic eye movements Working memory Visuospatial/visuomotor attention Updating spatial information (lateral) Temporal context recognition Same-different discrimination (right) Frequency deviant detection		Apraxia Deficits in contralateral fine motor control Difficulty in using sensory feedback Transient disturbance of the ability to initiate voluntary motor actions (SMA) Transient speech arrest (SMA)
17	Primary visual cortex (V1)	Striate cortex	Detection of light intensity, patterns Contour perception Colour discrimination Visual attention Visuo-spatial information processing (Right) Processing spatial orientation Tracking visual motion patterns Visual priming Horizontal saccadic eye movements		Blindsight Cortical blindness
18	Secondary visual cortex (V2, V3, V4, V5)	Middle occipital gyrus	V2	Detection of light intensity, patterns Feature-based attention	Visual agnosia Alexia
19		Inferior occipital gyrus	V3	Monochromatic pattern perception	Achromatopsia
			V4	Colour perception Shape perception	
			V5/MT	Movement detection Perception of direction Analysing stereoscopic depth	Akinetopsia
20	Inferior temporal, Fusiform and parahippocampal gyri		Visual fixation Integration of visual elements Dual working memory task processing		Agraphia
37	Posterior inferior temporal gyrus, middle temporal gyrus and fusiform gyrus		Episodic encoding Face recognition Visual motion processing Visual fixation Sustained attention to colour and shape Motion aftereffect		Prosopagnosia Visual agnosia Alexia
38	Temporal pole		Visual processing of emotional images Multimodal memory retrieval Response to tone stimulus Colour and structural judgments of familiar objects		Anomia Semantic memory impairment (left) Episodic memory impairment (right) High-level visual and auditory processing
46	Anterior middle frontal gyrus		Memory encoding and recognition Working memory Executive control of behaviour Horizontal saccadic eye movement		Sentence comprehension difficulties Memory impairment

SMA—Supplementary Motor Area, MT—middle temporal area.

**Table 2 biology-11-01720-t002:** The four typical situations of the signal detection theory.

		Response	
		Present (Yes)	Absent (No)
Signal	Present	Hit	Miss
	Absent	False alarm	Correct rejection

**Table 3 biology-11-01720-t003:** The effect of flickering light of different colours on the evoking of epileptic seizures. Adapted from: Parra et al. [111].

Frequency	Effect on Eliciting PPRs
10–15 Hz	Colour stimulation > white light
15 Hz	Red-blue altering stimulation
20–30 Hz	Colour stimulation = white stimulation Alternating colours are less provocative

## Data Availability

Not applicable.

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
