# Peer review of "Neuropsychological and Neurophysiological Mechanisms behind Flickering Light Stimulus Processing"

_biology, 2022, doi:10.3390/biology11121720_

Round 1
Reviewer 1 Report
The authors have done a great job at combining basic anatomical and physiological knowledge to provide background information on the visual system related to flickering. This is combined with many summaries of CFF publications. However, the manuscript misses explanations of why certain sections support the overall aim. The summarised literature is not integrated into the discussion, making it hard to follow the flow of information provided in the manuscript. Overall, the manuscript can be improved in structure and clearness.
Reviewer 2 Report
This manuscript reviewed current evidence on flickering light.
I think that this review was well written. However, there are too limited evidence on the effects of flickering light on Alzhemier's disease, while potential benefits on this disorder were too emphasized, which lead to misleading. It is necessary to revise the manuscript to avoid this misleading.
1) Please delete the sentences of ". Previous reports indicate that it can be used in the treatment of cognitive impairment, including those associated with Alzheimer's disease. " in the abstract.
2) In the introduction section, please delete the sentences of "Some studies have indicated the usefulness of flickering light in the detection of Alzheimer’s disease and the therapy of cognitive functions. It was shown that the dysfunction of memory is correlated with brainwave abnormalities (Locatelli et al., 1998; Miyauchi et al., 1994), but it could be enhanced by the 10 Hz flicker (Williams et al., 2006). Early detection and interventions in this kind of neurodegenerative disease could be beneficial to patients suffering from prolonged cognitive underperformance (Curran & Wattis, 2000). " That is because there are too limited, conflicting evidence of the effects of light on Alzhiemer's disease. It is necessary to avoid misleading and overestimation." Please rewrite the introduction without their potential benefits on cognitive disorders.
3) In the future direction section, please delete the sentences of "Since there have been reports that flickering light could potentially be used in the treatment of cognitive disorders and to support the functioning of patients with Alzheimer's disease, it is necessary to conduct the above-mentioned studies in the future to increase the range of possible applications. In one of the studies mentioned in this review (Williams et al., 2006), evidence was provided to show that memory impairment is correlated with brainwave abnormalities in EEG and that it can be reduced by flickering light at 10 Hz, so randomised clinical trials are needed to confirm these reports before implementing them in clinical practice." It is necessary to avoid misleading and overestimation on the effects of light on cognition. There are too limited evidence on the benefits of light on global cognition.
I think it is necessary to revise the manuscript.
Round 2
Reviewer 1 Report
The authors have improved the manuscript by providing introductions at the beginning of various sections. The final conclusions help to bind the whole manuscript together. However, the manuscript remains hard to read in certain parts as it is unclear what the point is that the authors are trying to convey.
1. In section 6 various neuroimaging studies are summarised, but no conclusion is draw from these studies. How are these studies relevant for this review at supporting the overall aim.
2. Figure 5 and 6 do not support the understanding of the section and can be removed.
3. Section 8 can use a conclusion at the end of the section.
4. Section 9 does not answer the question it starts with. Two studies are summarised, but no conclusion is drawn. The first study[99] has insignificant results. How does this section support the overall aim of the review?
5. Sections 10 does not support the aim of the review.
6. Section 11 mentions the use of BCI and VR as future options to combine with flickering light, but fails to deliver on how the combinations might benefit BCI /VR research.
Reviewer 2 Report
Thank you for revising the manuscript.
I think this manuscript would be suitable for publication.
Author Response
Dear Reviewer,
Thank you very much for your comments, your work, and your recommendation for the publication of this manuscript.
Round 3
Reviewer 1 Report
The authors, again, improved the manuscript. It is now a coherent story supported by literature and sound summaries and conclusion.